# Prevalence and Associated Factors of Clinical Myelopathy Signs in Smartphone-Using University Students with Neck Pain

**DOI:** 10.3390/ijerph19084890

**Published:** 2022-04-17

**Authors:** Rungthip Puntumetakul, Thiwaphon Chatprem, Pongsatorn Saiklang, Supaporn Phadungkit, Worawan Kamruecha, Surachai Sae-Jung

**Affiliations:** 1Research Center of Back, Neck, Other Joint Pain, and Human Performance (BNOJPH), Khon Kaen University, Khon Kaen 40002, Thailand; thiwaphon.ao@gmail.com; 2Department of Physical Therapy, Faculty of Associated Medical Sciences, Khon Kaen University, Khon Kaen 40002, Thailand; suppha@kku.ac.th (S.P.); workam@kku.ac.th (W.K.); 3Faculty of Physical Therapy, Srinakharinwirot University, Nakhon Nayok 26120, Thailand; pongsatornsa@g.swu.ac.th; 4Department of Orthopaedics, Faculty of Medicine, Khon Kaen University, Khon Kaen 40002, Thailand; sursea@kku.ac.th

**Keywords:** smartphone addiction, clinical myelopathic signs, university students, neck pain

## Abstract

University students have the highest smartphone-use addiction, which coincides with a rising number in instances of neck pain. As the time in smartphone use increases, neck flexion tends to increase. These positions can affect the spinal cord by the direct and indirect mechanisms which lead to cervical myelopathy. Thus, the current study aimed to determine the prevalence and associated factors of clinical myelopathic signs in smartphone-using university students with neck pain. A total of 237 smartphone-using university students with neck pain participated in the study. They were 20 to 25 years old. Their clinical myelopathic signs were evaluated using standardized test procedures. The prevalence of the clinical myelopathic sign was the Trömner sign at 41.35%, the finger escape sign at 28.27%, Hoffmann’s sign at 25.74%, and the inverted supinator sign at 18.14%. Smartphone usage ≥9.15 h per day was associated with ≥1 of a positive clinical myelopathic sign (adjusted OR = 1.85, 95% CI = 1.05 to 3.26, *p* = 0.05). The current study highlighted that prolonged smartphone usage may affect the spinal cord. Long duration (≥9 h per day) was associated with at least one positive clinical myelopathic sign. Therefore, smartphone-using university students need to keep their duration of smartphone use to less than 9 h per day. More attention should be given to increasing awareness about the importance of having healthy positions when using smartphones and using them for restricted durations in order to control the increasing prevalence of cervical myelopathy among smartphone-using university student in our societies.

## 1. Introduction

Smartphones can be used in many ways; as such, the smartphone is a device that essentially shrinks the world into our hands. Because of this convenience, people are becoming more dependent on their smartphones than ever before. The number of global smartphone users increased by 49.89% from 2017 to 2022, and 83.80% of the world’s population now own a smartphone [1]. However, along with the rise in smartphone use, potential risks for musculoskeletal problems have also been reported [2,3,4].

In the research on the prevalence of smartphone use, university students rank higher than any age group, and among them, a high rate of musculoskeletal problems has been reported, especially in the neck region [5,6]. Neck pain causes disabilities of varying severity and is responsible for substantial medical expenses and productivity losses [7]. Moreover, neck pain is a common symptom of cervical myelopathy [8,9,10], and up to 35% of cervical myelopathy cases diagnosed via magnetic resonance imaging (MRI) have been found among patients who currently have neck pain [11].

Cervical myelopathy is a neurological injury to the spinal cord caused by static and dynamic mechanisms [12,13,14] and other serious pathologies [15,16,17]. The diagnosis of cervical myelopathy can be difficult because the signs and symptoms vary widely among the population [9,10]. In an attempt to detect this condition for giving an appropriate intervention, diagnostic criteria were established. Cervical myelopathy is defined by gait dysfunction, loss of hand dexterity, and motor/sensory dysfunction corresponding to cervical cord compression in an MRI [9,10,18,19]. However, when these signs and symptoms are present, the provision of conservative treatment may be delayed as patients may be referred to surgical treatment instead [9].

A clinical myelopathy sign is a term used to describe the presence of Hoffmann’s sign, the inverted supinator sign [11,18,19,20], the Babinski sign [11,18,19,20,21,22], the Trömner sign [16], the finger escape sign [22,23], or the Clonus sign [10,20]. A study reported that clinical myelopathic signs were sensitive to the detection of cervical myelopathy [11,18,19,20,22,24,25,26]. Furthermore, research by Nagata et al. (2014) showed that clinical myelopathic signs are associated with a decrease in cervical spinal cord diameter (OR = 2.73, 95% CI = 1.83–4.23, *p* < 0.001) [27].

One case study on smartphone users and cervical myelopathy concluded that the onset of cervical myelopathy was related to maintaining a neck flexion posture overnight from falling asleep while using a smartphone. The authors explained that cervical myelopathy may have been due to spinal cord ischemia [28]. Hnin Lwin et al. reported that smartphone usage of ≥four h per day, an education at university or postgraduate level, and neck pain scores of >seven on a numerical rating scale are associated with clinical myelopathic signs with an adjusted OR (95% CI) of 2.57 (1.52 to 4.36), 3.35 (1.14 to 9.89), and 2.31 (1.07 to 4.99), respectively [26]. However, their study conducted on adults did not focus specifically on university students who had the highest rates of addiction to smartphone use [23].

A few studies have reported the prevalence of clinical myelopathy signs; one is hospital based [20] and another is community based [23], and all of them focus on those of adult age, including the elderly [20,26]. There is limited evidence of the prevalence and factors associated with clinical myelopathic signs in smartphone-using university students with neck pain. The current study aimed to investigate the prevalence of clinical myelopathic signs and to determine the factors associated with them among smartphone-using university students with neck pain.

## 2. Materials and Methods

### 2.1. Design

The study design was approved by the ethics review board of the Khon Kaen University Ethics Committee (HE632183). Before assessing the study participants, the investigators obtained their informed written consent.

### 2.2. Participants

Between July 2020 and October 2020, smartphone-using university students with neck pain were recruited from Khon Kaen University, Khon Kaen, Thailand, via a social media advertisement. Participants were eligible for inclusion if they (i) were university students, (ii) had reported neck pain for more than three months [29], (iii) had reported daily smartphone use of at least two hours per day for more than six months [30], and (iv) had a pain score between 3–7 on the visual analog scale (VAS) [31]. Participants were excluded if they (i) had a positive jaw jerk; (ii) had a positive Spurling test; (iii) had a history of previous cervical spine surgery; (iv) were concurrently suffering from other locomotor disorders; (v) had a history of brain trauma; (vi) had comorbid neurological diseases, such as cerebral infarction or neuropathy; (vii) had consumed any sedative drugs or alcohol within the past 48 h; or (viii) were pregnant [26].

After completing the recruitment process, participants were asked questions to obtain (i) demographic data (e.g., age, sex, weight, height), (ii) smartphone usage data, and (iii) computer or laptop usage data. This process was conducted by the researcher, PS.

### 2.3. Sample Size Determination

The sample size was calculated according to the prevalence of clinical myelopathic sign(s) as the primary outcome. The highest reported prevalence (inverted supinator sign) was 0.19 from Rhee et al. (2009) [20]. The sample size was calculated according to the formula $n={\left(Z\frac{\alpha }{2}\right)}^{2}P\left(1-P\right)/e^{2}$, where we set $Z\frac{\alpha }{2}=1.96$ and e = 0.05. At least 237 participants were required to participate in the study.

### 2.4. Clinical Myelopathic Signs Examination

Clinical myelopathic signs were observed randomly by the researcher, RP, who had 30 years of clinical experience. These signs included (i) Hoffman’s sign, (ii) the Trömner sign, (iii) the inverted supinator sign, (iv) the finger escape sign, (v) the Babinski sign, and (vi) the Clonus sign. We assessed the participants and controls in seated (Hoffman’s sign, the Trömner sign, the inverted supinator sign, and the finger escape sign) and lying positions (the Babinski sign and the Clonus sign). In this way, we could minimize the influence of excitability on other muscles.

Hoffmann’s sign was performed by supporting the participant’s hand in a relaxed wrist dorsiflexion position with the finger partially flexed. The middle finger was firmly grasped and partially extended, and the nail was snapped by the examiner’s thumbnail. A positive Hoffmann’s sign was considered present if flexion of the distal interphalangeal joint was noted on either the thumb or index finger [19,32].

The inverted supinator sign test was performed by supporting the participant’s forearm in a neutral rotation so that the radial aspect of the forearm was facing upward, and the wrist was in an ulnar deviation. The radial aspect of the forearm was then tapped 6 cm from the radial styloid process. A positive sign was indicated by involuntary flexion of the four-finger flexors with a diminished brachioradialis reflex [19,20].

For the finger escape sign, a pillow was placed on the participants’ lap to support their elbow, which was positioned at 90°. The forearms were pronated, and all fingers were adducted and placed on the pillow. Participants were asked to adduct their fingers for 30 s. The positive finding was the participant’s inability to maintain adduction of the fifth digit, which would have drifted in an ulnar and volar direction [18].

The Trömner sign, an alternative test designed to provide similar information to Hoffmann’s sign, was performed by tapping or flicking the volar surface of the distal phalanx of the middle finger, which was held in a partially flexed position between the examiner’s finger and thumb. A positive response was flexion of the thumb and/or index finger [19,33].

The Clonus sign was elicited by forcefully dorsiflexing the ankle and maintaining pressure on the sole while observing rhythmic beats of ankle flexion and extension as a positive result [20].

The Babinski sign was elicited by rubbing the lateral border of the foot using the blunt end of a hammer jerk in a proximal-to-distal direction and then medially across the ball of the foot. A positive sign was defined as a hallux position, which is essentially dorsiflexion, with the lesser toes fanning out [19].

All participants were assessed using six clinical myelopathic signs. If the participants had at least one sign positive, they were categorized into “≥1 positive clinical myelopathic sign(s)” group. In contrast, participants who did not display any positive clinical myelopathic signs were included in the “no positive clinical myelopathic sign” group.

### 2.5. Statistical Analysis

Descriptive statistics were used to analyze the participants’ characteristics and clinical myelopathic signs. Continuous variables were analyzed by the mean and standard deviation (SD). Categorical variables were considered in terms of frequency and percentage. Each variable was categorized into an ordinal scale for further logistic regression analysis.

Univariate logistic regression was used to determine the association between factors and a participant’s positive, at least one, clinical myelopathic sign. The variables that reached a *p*-value of less than 0.2 in the univariate logistic regression analysis were included in the multiple logistic regression model. The backward stepwise elimination process was applied for multivariate logistic regression analysis; variables with *p*-values of less than 0.05 were considered statistically significant. The data were analyzed using the STATA program version 10 (STATA, College Station, TX, USA).

## 3. Results

Of the 240 participants recruited for eligibility, three reported daily use of smartphones for less than two h per day. Thus, a total of 237 smartphone-using university students with neck pain were included in this study. General participant characteristics are presented in Table 1.

The participants’ average age was 20.54 ± 1.35 years. There were 142 (59.92%) males and 95 (40.08%) females. Most of the participants had a normal body mass index; BMI (<25 kg/m^2^). They used computers or laptops and smartphones for an average of 5.58 ± 3.94 h and 9.15 ± 3.98 h per day, respectively. The start time until the current time of computer or laptop and smartphone were 6.45 ± 3.59 years and 8.62 ± 2.36 years. The participants frequently adopted a sitting posture (42.65%) when using their smartphones.

The number of positive clinical myelopathic signs in this group of smartphone users is shown in Figure 1. The prevalence of these signs from high to low was the Trömner sign at 41.35%, the finger escape sign at 28.27%, Hoffmann’s sign at 25.74%, the inverted supinator sign at 18.14%, the Babinski sign at 0%, and the Clonus sign at 0%. When grouping the positive clinical myelopathic signs, as shown in Table 2, it was revealed that a rate of more than one positive clinical myelopathic sign was the most prevalent.

Table 3 shows the crude odds ratio for the univariate logistic regression between at least one positive clinical myelopathic sign and its associated factors. They were (i) high ≥ 164.4 cm, (ii) personal computer or laptop usage of ≥5.58 h per day, (iii) smartphone usage of ≥9.15 h per day, and (iv) the start time until the current time of smartphone use ≥ 8.62 in years. These factors were then analyzed using multivariate analysis. The results showed that only smartphone usage of ≥9.15 h per day was associated with at least one positive clinical myelopathic sign.

## 4. Discussion

The purpose of the present study was to investigate the prevalence and associated factors of clinical myelopathic signs in smartphone-using university students with neck pain. This study adopted clinical myelopathic signs that are commonly used in general clinical settings because of their sensitivity and convenience [20]. The prevalence of these signs from high to low was the Trömner sign at 41.35%, the finger escape sign at 28.27%, Hoffmann’s sign at 25.74%, the inverted supinator sign at 18.14%, the Babinski sign at 0%, and the Clonus sign at 0%.

The Trömner sign was found to have the highest prevalence (up to 41.35%) in the current study, which was consistent with the findings of Chiyamongkol et al.’s study [19]. Chiyamongkol et al. reported a positive Trömner sign rate of 41.18% when considered together with a cord compression sign on an MRI. They also found a high sensitivity (94%) for the Trömner sign in detecting even a mild degree of cervical myelopathy. In Chang et al.’s study, a positive correlation was found between the amplitude of muscle action potential obtained with the Trömner sign and the cord compression ratio [34]. Therefore, we hypothesized that our participants may have had a mild degree of cervical myelopathy.

The prevalence of finger escape signs was 28.27%, which conflicts with the study of Hnin Lwin et al. [26], who reported the prevalence at 10% [26]. This difference may be because the current study included younger participants. Those of a younger age may have a higher response to nerve stimulation than those of an older age [27,35]. In contrast, Wang et al. showed the prevalence of finger escape signs to be up to 55% [23]. The higher prevalence in their study may be because all the participants were recruited from a hospital-based setting, and most had been diagnosed with cervical myelopathy by MRI [23]. Another study showed that 75% of cervical myelopathic patients have decreased dexterity in the intrinsic muscles of the hand, leading to problems in handwriting, typing, shirt buttoning, or other motor track systems [10]. Finger escape signs may, therefore, be used for early detection before other hand problems are found.

Hoffmann’s sign was observed at a prevalence of 25.74%, which was similar to the results of Chiyamongkol et al.’s study [19]. Chiyamongkol et al. reported a prevalence of 38.82% and a sensitivity rate of 76% for Hoffman’s sign [19]. In contrast, Nagata et al. reported a 1.7% prevalence of Hoffmann’s sign in the general population, which was significantly lower than the results of this study [27]. This difference can be attributed to Nagata et al. having conducted their study in an older normal population, with an average age of 67 years. In older adults, exaggerated reflexes are uncommon, as reflexes may be reduced by peripheral neuropathy or other causes [27]. Therefore, age is important, as different age groups have a different prevalence of myelopathic signs.

The prevalence of inverted supinator reflexes in the present study was 18.14%. This result is consistent with the prevalence reported by Rhee et al. and Hnin Lwin et al. at 19% [20] and 20.36% [26], respectively. These studies involved patients with neck pain, as in our study. However, the setting was different; the study by Rhee et al. was hospital based [20]. Our study and Hnin Lwin et al.’s study were community based, in that the prevalence of the inverted supinator reflex is comparable to the hospital base, so this test may have a high sensitivity to cervical myelopathy even in a general individual with neck pain. The inverted supinator reflex represents cervical cord problems, especially at the C5–C6 level [10]. Therefore, smartphone-using university students with neck pain may have cervical spinal cord problems, particularly at the C5–C6 level.

The Babinski and Clonus signs were both observed at rates of 0%, which was similar to the findings of Rhee et al.’s and Hnin Lwin et al.’s studies [20,26], where corresponding rates of 0% and 3% were reported [20,26]. The Babinski sign is known to be a primitive reflex present in the immature nervous system and reflects more severe corticospinal tract impairments than other clinical myelopathic signs. For this reason, the Babinski and Clonus signs are thought to be nearly always associated with a true upper motor neuron lesion, and false-positive results are rare [32]. However, this study investigated the irritation of the spinal cord via the assessment of clinical myelopathic signs; our results revealed that no participants exhibited the Babinski or Clonus signs.

Cook et al. (2010) reported that exhibiting at least one out of five clinical myelopathy signs (gait deviation, Hoffmann’s sign, the inverted supinator sign, the Babinski sign, and age > 45 years) had the highest sensitivity (94%) for ruling out cervical myelopathy among patients with neck pain [36]. This was consistent with the findings of the present study, which showed that exhibiting at least one clinical myelopathy sign had the highest prevalence (32.07%) of the other two to six positive clinical myelopathic signs. Therefore, only one sign may be clinically useful for ruling out patients with cervical myelopathy.

In the present study, smartphone usage of ≥nine h per day was associated with ≥one positive clinical myelopathic sign, with an adjusted odds ratio of 1.85 (95% CI = 1.05 to 3.26). Previous studies have found that smartphone use leads to musculoskeletal problems, particularly in the neck region [3,4,5], and that the neck is most affected when using smartphones in the sitting posture [5,26,37]. In our study, approximately 42% of the participants reported using their smartphones in a sitting position. When sitting for long periods of time, smartphone users tend to develop an increased neck flexion posture [38,39,40,41].

Since smartphone usage induces more neck flexion than the use of other visual display terminals [39,40], prolonged smartphone usage can increase mechanical stress on cervical spine structures [6,42]. This may lead to ligamentous edema in the cervical spine. Hypertrophic changes in the ligament can cause a decrease in the sagittal diameter of the spinal canal. Neck flexion causes lengthening of the cervical spinal cord; it also causes narrowing of the anteroposterior cord diameter [43]. Narrow anteroposterior spinal cord diameter during neck flexion, together with ligamentous edema, may be essential to the development of myelopathy [12,44,45]. Additionally, changes in neck flexion/extension, which narrow the cervical spinal canal dynamically, place more increased strain and shear forces on the spinal cord [12,46].

For the flexion position of the cervical spine, the length of the spinal cord was increased, and there was stretching of the cord and dura [47]. The cord was stretched in proportion to the amount of flexion, being greatest in the lower cervical segment [43]. Stretch might be a potent factor in increasing the damage of any pathological process within the cord substance by disrupting the relationships between nerve cells and fibers and their delicate vessels [47] as this permits neurological injury [12,13]. Therefore, not only may movements and pressure inflict direct trauma on the cord, but, by interference with its blood supply, they may conceivably cause indirect damage.

The present study has some limitations. First, we did not measure the cervical flexion angle or weight of the smartphones. Second, MRI is the standard investigation of cervical myelopathy, but it was not used in the current study. Lastly, the cross-sectional study design has limited the investigation of causality between the risk factors and neck pain among university students who are smartphone users with positive clinical myelopathic signs. A prospective study is underway to follow up with all respondents so that the causal relationship between prognostic/risk factors and neck pain among university students with smartphones who have positive clinical myelopathic signs can be explored. Other possible factors, such as sports or fitness activities and the psychosocial characteristics of neck pain, may influence our findings. Therefore, further study is needed to investigate these factors in more detail.

## 5. Conclusions

This is the first study to investigate the prevalence and relationship between risk factors and positive clinical myelopathic signs in smartphone-using university students with neck pain.

In this study, the Trömner sign was the most common among the participants. Furthermore, we found that a daily smartphone usage rate of ≥nine h is associated with a nearly twofold increase in the chance of exhibiting at least one positive clinical myelopathic sign among university students with neck pain, which may be a warning sign of cervical myelopathy. More attention should be directed to increasing awareness about the importance of maintaining a healthy posture when using a smartphone and limiting use to restricted durations to control the increasing prevalence of cervical myelopathy among university-aged smartphone users in our societies.

## Figures and Tables

**Figure 1 ijerph-19-04890-f001:**
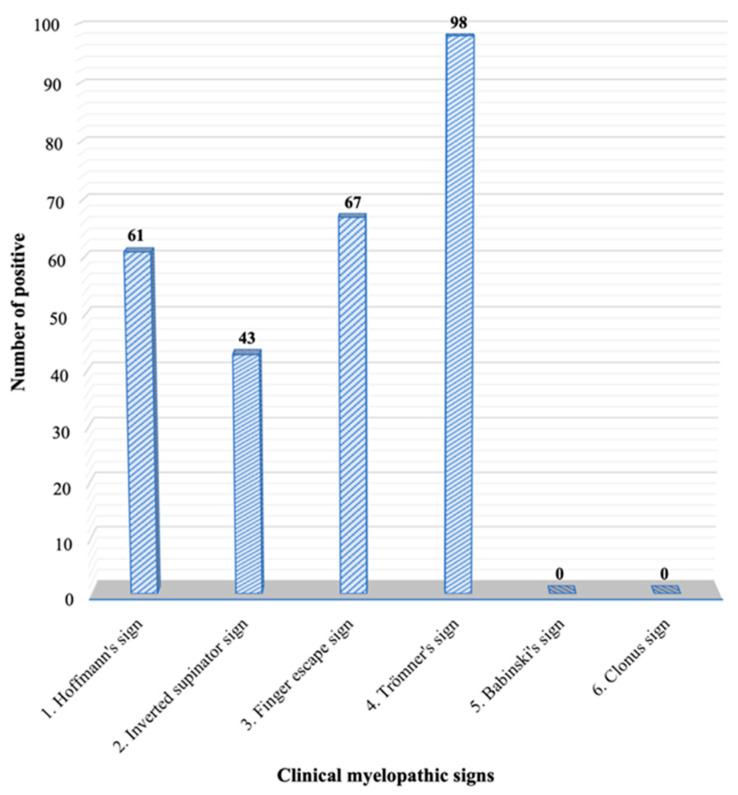
The number of each positive clinical myelopathic sign.

**Table 1 ijerph-19-04890-t001:** Demographic information of smartphone-using university students with neck pain (*n* = 237).

Characteristics	N (%)	Mean ± SD	Range
Age (years)		20.54 ± 1.35	19 to 25
Sex			
Female	95 (40.08)		
Male	142 (59.92)		
Weight (kg)		58.94 ± 14.57	39 to 120
Height (cm)		164.49 ± 8.73	148 to 185
Body Mass Index (kg/m^2^)			
Normal (<25 kg/m^2^)	194 (81.86)		
Overweight (≥25 kg/m^2^)	43 (18.14)		
Personal computer or laptop usage hours per day (hours)		5.58 ± 3.94	0 to 20
The start time until the current time of the personal computer or laptop usage (years)		6.45 ± 3.95	0 to 20
Smartphone usage hours per day (hours)		9.15 ± 3.98	2 to 20
The start time until the current time of smartphone usage (years)		8.62 ± 2.36	4 to 18
Posture in use smartphone			
Sitting	101 (42.62)		
Lying	78 (32.91)		
Standing	2 (0.84)		
Sitting and lying	38 (16.03)		
Sitting, lying, and standing	18 (7.60)		

Abbreviation: SD, Standard deviation.

**Table 2 ijerph-19-04890-t002:** Group of positive clinical myelopathic signs in smartphone-using university students with neck pain (*n* = 237).

Group of Positive Clinical Myelopathic Signs	N (%)
No positive clinical myelopathic sign	83 (35.02)
1 positive clinical myelopathic sign	76 (32.07)
2 positive clinical myelopathic signs	51 (21.52)
3 positive clinical myelopathic signs	23 (9.70)
4 positive clinical myelopathic signs	4 (1.69)

**Table 3 ijerph-19-04890-t003:** The relationship between smartphone-using university students with neck pain who have a positive clinical myelopathic sign and association factors (*n* = 237).

Characteristics	≥1 of Positive Clinical Myelopathic Sign	
	OR Crude (95% CI)	OR Adjust (95% CI)
Age (years)		
<20.58 ± 1.35	1	-
≥20.58 ± 1.35	1.17 (0.68 to 2.01)	
Sex		
Male	1	-
Female	0.75 (0.43 to 1.30)	
Weight (kg)		
<58.94 ± 14.57	1	-
≥58.94 ± 14.57	0.95 (0.59 to 1.64)	
Height (cm)		
<164.49 ± 8.73	1	1
≥164.49 ± 8.73	0.68 (0.40 to 1.17) *	0.73 (0.42 to 1.27)
Body Mass Index (kg/m^2^)		
Normal (<25 kg/m^2^)	1	-
Overweight (≥25 kg/m^2^)	1.19 (0.58 to 2.44)	
Personal computer or laptop usage hours per day (hours)		
<5.58 ± 3.94	1	-
≥5.58 ± 3.94	1.47 (0.85 to 2.55) *	
The start time until the current time of the personal computer or laptop usage (years)		
<6.45 ± 3.95	1	-
≥6.45 ± 3.95	0.82 (0.48 to 1.40)	
Smartphone usage hours per day (hours)		
<9.15 ± 3.98	1	1
≥9.15 ± 3.98	1.82 (1.04 to 3.18) *	1.85 (1.05 to 3.26) **
The start time until the current time of smartphone usage (years)		
<8.62 ± 2.36	1	1
≥8.62 ± 2.36	1.43 (0.83 to 2.47) *	1.50 (0.86 to 2.64)
Posture in use smartphone		
Other	1	-
Sitting	1.05 (0.61 to 1.82)	

Note: * Significant at the *p*-value < 0.2 level was included into the model of logistic regression, ** Significant at the *p*-value < 0.05 level.

## Data Availability

The data will be available for anyone who wishes to access them for research purpose and contract should be made via the corresponding author rungthiprt@gmail.com.

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
