# Peer review of "Prevalence and Associated Factors of Clinical Myelopathy Signs in Smartphone-Using University Students with Neck Pain"

_ijerph, 2022, doi:10.3390/ijerph19084890_

Round 1
Reviewer 1 Report
Dear authors, it is excellent to go through your study. You have find out on of the causative factor for cervical myelopathy is prolonged uses of smartphone by University students. At era of technology, its use have sever adverse effects on human health especially in young, dynamic University students, topic is timely addressed.
There some studies recently done on smartphone uses and its effects on cervical disorders, it will be great to cites these studies.
- Adel Alshahrani , Mohamed Samy Abdrabo, Sobhy M. Aly, Mastour Saeed Alshahrani, Raee S. Alqhtani, Faisal Asiri, and Irshad Ahmad - Effect of Smartphone Usage on Neck Muscle Endurance, Hand Grip and Pinch Strength among Healthy College Students: A Cross-Sectional Study. Int. J. Environ. Res. Public Health 2021, 18, 6290. https://doi.org/10.3390/ijerph18126290.
Author Response
The authors appreciate the feedback from the reviewer for your precious time in reviewing our manuscript. We have responded to your comments on a point-by-point basis. The revised version of the manuscript is marked with a yellow highlight changed for reviewer 1. We have carefully checked the format for revised manuscripts in submission guidelines and sent it for English editing.

Reviewer 2 Report
Puntumetakul_Myelopathy_IJERPH_2022
Reviewer report
Thank you for presenting your manuscript. You have a well-designed study, the work was carefully executed, but I have some concerns highlighted bellow.
General comments
Extensive English edition is recommended.
Specific comments
Introduction
The article includes a comprehensive introduction and background. This section is sufficient to demonstrate the justification for the development of the study in the field of knowledge.
Materials and Methods
2.2. Participants. How was the recruitment process performed?
2.3. Sample size determination. Please determine the meaning of 0.19 (P-value). I suppose that it is 19% (Prevalence/Proportion-value?).
2.5. Statistical analysis. Could you please explain a little more the way the logistic regression models were performed? My principal concern is the way the positive clinical myelopathic signs were used. I understand that they were the dependent variable, but, one Univariate logistic regression and one multivariate logistic regression analysis were performed for each clinical myelopathic sign?
But in that case, How can we interpret table 3 in the results section?
I understand that OR Crude refers to the univariate logistic regression and OR adjust to the multivariate logistic regression, but in that case, to which clinical myelopathic sign refers the OR? The sentence “Any one of clinical myelopathic sign positive”, that it is explaining the columns, it is not clear. After reading discussion section it looks like that the variable clinical myelopathic sign positive has been recodifies in two categories: No clinical myelopathic sign positive/ >1 of any clinical myelopathic sign positive, but all these aspects must be clarified in material and methods, and in the results sections.
- Results
Line 150. “Of the 240 included participants”, please change to: “Of the 240 participants recruited for eligibility”.
Line 200. Please, eliminate this sentence: “Therefore, our participant's neck pain university with smartphone users may have a mild degree of cervical myelopathy.” I think that with the data of the current study it is not possible to make this affirmation.
- Discussion
Line 250. “In this study, smartphone usage of ≥9 hours per day was associated with ≥1 clinical myelopathic sign positive with an adjusted odds ratio of 1.85 (95%CI=1.25 to 3.26).” I think there is an error: in the table 3: 1.85 (1.05 to 3.26).
Line 275. “As mentioned above, a university student with uses a daily smartphone for more than 9 hours has nearly 2 times developed cervical myelopathy which was at least one clinical myelopathic sign positive.” Please change to: “As mentioned above, a university student with uses a daily smartphone for more than 9 hours has nearly 2 times to developed at least one clinical myelopathic sign positive.”
Line 278. Limitations. Please, add a limitation in relation to the absence of imaging tests to contrast the diagnosis of myelopathy.
- Conclusions
Line 290. “A daily ≥9 hour of smartphone usage is associated with a nearly 2 times increase in odds of experiencing cervical myelopathy among neck pain university students.”
Please, change to: “A daily ≥9 hour of smartphone usage is associated with a nearly 2 times increase in odds of experiencing at least one clinical myelopathic sign positive among neck pain university students.”
Author Response
The authors appreciate the feedback from the reviewer for your precious time in reviewing our manuscript. We have responded to your comments on a point-by-point basis. The revised version of the manuscript is marked with a grey highlight change for reviewer 1. We have carefully checked the format for revised manuscripts in submission guidelines and sent it for English editing.

Reviewer 3 Report
Dear all
I realize that authors have many journals to consider when they want to publish their work, so I appreciate your interest in "Int. J. Environ. Res. Public Health"; I am very happy to be able to write in a positive way. It is evident that you have put a great deal of effort into this project and I want to praise your efforts, Fortunately, the actual contribution from your study is clear and strong, the manuscript as currently written suggests that it might be suitable for sharing information about this topic, but the manuscript that you reported, needs few minor edits. I should like to thank you for give me an opportunity to consider this work for publication. It may be that the you would like to consider resubmitting it, in which case I hope that the comments from my review may help you to revise it before resubmitting it. These comments are given below.
Best Regards
- Title: I suggest changing the title with Prevalence and Association Factors of Clinical Myelopathy Signs in University Students Smartphone Users with Neck Pain;
- Introduction section: references are missing in the few sentences; neck pain and myelopathy can also be caused by serious pathologies; insert a part relating to the differential diagnosis and the attention that must be paid to the evaluation of red flags, within the physical examination and clinical history of patients such as the evaluation of cranial nerves, vascular problems, rare cranio-cervical malformations. I suggest adding references: - A guide to cranial nerve testing for musculoskeletal clinicians. J Man Manip Ther. 2021 Dec;29(6):376-389. doi: 10.1080/10669817.2021.1937813. Epub 2021 Jun 29.PMID: 34182898; -Assessing Vascular Function in Patients With Neck Pain, Headache, and/or Orofacial Pain: Part of the Job Description of All Physical Therapists. J Orthop Sports Phys Ther. 2021 Sep;51(9):418-421. doi: 10.2519/jospt.2021.10408. Epub 2021 May 10.PMID: 33971733; - Basilar impression presenting as intermittent mechanical neck pain: a rare case report. BMC Musculoskelet Disord. 2016 Jan 11;17:7. doi: 10.1186/s12891-015-0847-0.PMID: 26754441;
-
in Materials and Methods: lack information in Patient characteristics, about smoking habits, alcohol, sports-fitness activity performed, previous history of neck pain, psychosocial characteristics such as fears of movement, worries about the study load, etc.
- in discussion section: Discussions should be reviewed, redundant sentences and prewritten information should be avoided. Focus on take-home messages and how that information impacts the clinical practice of management these patients
Author Response
The authors appreciate the feedback from the reviewer for your precious time in reviewing our manuscript. We have responded to your comments on a point-by-point basis. The revised version of the manuscript is marked with a green highlight changed for reviewer 3. We have carefully checked the format for revised manuscripts in submission guidelines and sent it for English editing.
